# EX$^2$: Exploration with Exemplar Models for Deep Reinforcement Learning

**Justin Fu**$^*$    **John D. Co-Reyes**$^*$    **Sergey Levine**
University of California Berkeley
{justinfu,jcoreyes,svlevine}@eecs.berkeley.edu

## Abstract

Deep reinforcement learning algorithms have been shown to learn complex tasks using highly general policy classes. However, sparse reward problems remain a significant challenge. Exploration methods based on novelty detection have been particularly successful in such settings but typically require generative or predictive models of the observations, which can be difficult to train when the observations are very high-dimensional and complex, as in the case of raw images. We propose a novelty detection algorithm for exploration that is based entirely on discriminatively trained exemplar models, where classifiers are trained to discriminate each visited state against all others. Intuitively, novel states are easier to distinguish against other states seen during training. We show that this kind of discriminative modeling corresponds to implicit density estimation, and that it can be combined with count-based exploration to produce competitive results on a range of popular benchmark tasks, including state-of-the-art results on challenging egocentric observations in the vizDoom benchmark.

## 1  Introduction

Recent work has shown that methods that combine reinforcement learning with rich function approximators, such as deep neural networks, can solve a range of complex tasks, from playing Atari games (Mnih et al., 2015) to controlling simulated robots (Schulman et al., 2015). Although deep reinforcement learning methods allow for complex policy representations, they do not by themselves solve the exploration problem: when the reward signals are rare and sparse, such methods can struggle to acquire meaningful policies. Standard exploration strategies, such as $\epsilon$-greedy strategies (Mnih et al., 2015) or Gaussian noise (Lillicrap et al., 2015), are undirected and do not explicitly seek out interesting states. A promising avenue for more directed exploration is to explicitly estimate the novelty of a state, using predictive models that generate future states (Schmidhuber, 1990; Stadie et al., 2015; Achiam & Sastry, 2017) or model state densities (Bellemare et al., 2016; Tang et al., 2017; Abel et al., 2016). Related concepts such as count-based bonuses have been shown to provide substantial speedups in classic reinforcement learning (Strehl & Littman, 2009; Kolter & Ng, 2009), and several recent works have proposed information-theoretic or probabilistic approaches to exploration based on this idea (Houthooft et al., 2016; Chentanez et al., 2005) by drawing on formal results in simpler discrete or linear systems (Bubeck & Cesa-Bianchi, 2012). However, most novelty estimation methods rely on building generative or predictive models that explicitly model the distribution over the current or next observation. When the observations are complex and high-dimensional, such as in the case of raw images, these models can be difficult to train, since generating and predicting images and other high-dimensional objects is still an open problem, despite recent progress (Salimans et al., 2016). Though successful results with generative novelty models have been reported with simple synthetic images, such as in Atari games (Bellemare et al., 2016; Tang et al., 2017), we show in our

---

$^*$equal contribution.

experiments that such generative methods struggle with more complex and naturalistic observations, such as the ego-centric image observations in the vizDoom benchmark.

How can we estimate the novelty of visited states, and thereby provide an intrinsic motivation signal for reinforcement learning, without explicitly building generative or predictive models of the state or observation? The key idea in our $EX^2$ algorithm is to estimate novelty by considering how easy it is for a discriminatively trained classifier to distinguish a given state from other states seen previously. The intuition is that, if a state is easy to distinguish from other states, it is likely to be novel. To this end, we propose to train *exemplar models* for each state that distinguish that state from all other observed states. We present two key technical contributions that make this into a practical exploration method. First, we describe how discriminatively trained exemplar models can be used for implicit density estimation, allowing us to unify this intuition with the theoretically rigorous framework of count-based exploration. Our experiments illustrate that, in simple domains, the implicitly estimated densities provide good estimates of the underlying state densities without any explicit generative training. Second, we show how to amortize the training of exemplar models to prevent the total number of classifiers from growing with the number of states, making the approach practical and scalable. Since our method does not require any explicit generative modeling, we can use it on a range of complex image-based tasks, including Atari games and the vizDoom benchmark, which has complex 3D visuals and extensive camera motion due to the egocentric viewpoint. Our results show that $EX^2$ matches the performance of generative novelty-based exploration methods on simpler tasks, such as continuous control benchmarks and Atari, and greatly exceeds their performance on the complex vizDoom domain, indicating the value of implicit density estimation over explicit generative modeling for intrinsic motivation.

## 2    Related Work

In finite MDPs, exploration algorithms such as $E^3$ (Kearns & Singh, 2002) and R-max (Brafman & Tennenholtz, 2002) offer theoretical optimality guarantees. However, these methods typically require maintaining state-action visitation counts, which can make extending them to high dimensional and/or continuous states very challenging. Exploring in such state spaces has typically involved strategies such as introducing distance metrics over the state space (Pazis & Parr, 2013; Kakade et al., 2003), and approximating the quantities used in classical exploration methods. Prior works have employed approximations for the state-visitation count (Tang et al., 2017; Bellemare et al., 2016; Abel et al., 2016), information gain, or prediction error based on a learned dynamics model (Houthooft et al., 2016; Stadie et al., 2015; Achiam & Sastry, 2017). Bellemare et al. (2016) show that count-based methods in some sense bound the bonuses produced by exploration incentives based on *intrinsic motivation*, such as model uncertainty or information gain, making count-based or density-based bonuses an appealing and simple option.

Other methods avoid tackling the exploration problem directly and use randomness over model parameters to encourage novel behavior (Chapelle & Li, 2011). For example, bootstrapped DQN (Osband et al., 2016) avoids the need to construct a generative model of the state by instead training multiple, randomized value functions and performs exploration by sampling a value function, and executing the greedy policy with respect to the value function. While such methods scale to complex state spaces as well as standard deep RL algorithms, they do not provide explicit novelty-seeking behavior, but rather a more structured random exploration behavior.

Another direction explored in prior work is to examine exploration in the context of hierarchical models. An agent that can take temporally extended actions represented as action primitives or skills can more easily explore the environment (Stolle & Precup, 2002). Hierarchical reinforcement learning has traditionally tried to exploit temporal abstraction (Barto & Mahadevan, 2003) and relied on semi-Markov decision processes. A few recent works in deep RL have used hierarchies to explore in sparse reward environments (Florensa et al., 2017; Heess et al., 2016). However, learning a hierarchy is difficult and has generally required curriculum learning or manually designed subgoals (Kulkarni et al., 2016). In this work, we discuss a general exploration strategy that is independent of the design of the policy and applicable to any architecture, though our experiments focus specifically on deep reinforcement learning scenarios, including image-based navigation, where the state representation is not conducive to simple count-based metrics or generative models.

Concurrently with this work, Pathak et al. (2017) proposed to use discriminatively trained exploration bonuses by learning state features which are trained to predict the action from state transition pairs. Then given a state and action, their model predicts the features of the next state and the bonus is calculated from the prediction error. In contrast to our method, this concurrent work does not attempt to provide a probabilistic model of novelty and does not perform any sort of implicit density estimation. Since their method learns an inverse dynamics model, it does not provide for any mechanism to handle novel events that do not correlate with the agent's actions, though it does succeed in avoiding the need for generative modeling.

## 3   Preliminaries

In this paper, we consider a Markov decision process (MDP), defined by the tuple $(\mathcal{S}, \mathcal{A}, \mathcal{T}, R, \gamma, \rho_0)$. $\mathcal{S}, \mathcal{A}$ are the state and action spaces, respectively. The transition distribution $\mathcal{T}(s'|a, s)$, initial state distribution $\rho_0(s)$, and reward function $R(s, a)$ are unknown in the reinforcement learning (RL) setting and can only be queried through interaction with the MDP. The goal of reinforcement learning is to find the optimal policy $\pi^*$ that maximizes the expected sum of discounted rewards, $\pi^* = \arg\max_\pi E_{\tau \sim \pi}[\sum_{t=0}^{T} \gamma^t R(s_t, a_t)]$, where, $\tau$ denotes a trajectory $(s_0, a_0, ...s_T, a_T)$ and $\pi(\tau) = \rho_0(s_0) \prod_{t=0}^{T} \pi(a_t|s_t)T(s_{t+1}|s_t, a_t)$. Our experiments evaluate episodic tasks with a policy gradient RL algorithm, though extensions to infinite horizon settings or other algorithms, such as Q-learning and actor-critic, are straightforward.

Count-based exploration algorithms maintain a state-action visitation count $N(s, a)$, and encourage the agent to visit rarely seen states, operating on the principle of optimism under uncertainty. This is typically achieved by adding a reward bonus for visiting rare states. For example, MBIE-EB (Strehl & Littman, 2009) uses a bonus of $\beta/\sqrt{N(s, a)}$, where $\beta$ is a constant, and BEB (Kolter & Ng, 2009) uses a $\beta/(N(s, a) + |\mathcal{S}|)$. In the finite state and action spaces, these methods are PAC-MDP (for MBIE-EB) or PAC-BAMDP (for BEB), roughly meaning that the agent acts suboptimally for only a polynomial number of steps. In domains where explicit counting is impractical, pseudo-counts can be used based on a density estimate $p(s, a)$, which typically is done using some sort of generatively trained density estimation model (Bellemare et al., 2016). We will describe how we can estimate densities using only discriminatively trained classifiers, followed by a discussion of how this implicit estimator can be incorporated into a pseudo-count novelty bonus method.

## 4   Exemplar Models and Density Estimation

We begin by describing our discriminative model used to predict novelty of states visited during training. We highlight a connection between this particular form of discriminative model and density estimation, and in Section 5 describe how to use this model to generate reward bonuses.

### 4.1   Exemplar Models

To avoid the need for explicit generative models, our novelty estimation method uses *exemplar models*. Given a dataset $X = \{x_1, ...x_n\}$, an exemplar model consists of a set of $n$ classifiers or discriminators $\{D_{x_1}, ....D_{x_n}\}$, one for each data point. Each individual discriminator $D_{x_i}$ is trained to distinguish a single positive data point $x_i$, the "exemplar," from the other points in the dataset $X$. We borrow the term "exemplar model" from Malisiewicz et al. (2011), which coined the term "exemplar SVM" to refer to a particular linear model trained to classify each instance against all others. However, to our knowledge, our work is the first to apply this idea to exploration for reinforcement learning. In practice, we avoid the need to train $n$ distinct classifiers by amortizing through a single exemplar-conditioned network, as discussed in Section 6.

Let $P_\mathcal{X}(x)$ denote the data distribution over $\mathcal{X}$, and let $D_{x^*}(x) : \mathcal{X} \to [0, 1]$ denote the discriminator associated with exemplar $x^*$. In order to obtain correct density estimates, as discussed in the next section, we present each discriminator with a balanced dataset, where half of the data consists of the exemplar $x^*$ and half comes from the background distribution $P_\mathcal{X}(x)$. Each discriminator is then trained to model a Bernoulli distribution $D_{x^*}(x) = P(x = x^*|x)$ via maximum likelihood. Note that the label $x = x^*$ is noisy because data that is extremely similar or identical to $x^*$ may also occur in the background distribution $P_\mathcal{X}(x)$, so the classifier does not always output 1. To obtain the

maximum likelihood solution, the discriminator is trained to optimize the following cross-entropy objective

$$D_{x^*} = \arg\max_{D \in \mathcal{D}} \left( E_{\delta_{x^*}}[\log D(x)] + E_{P_{\mathcal{X}}}[\log 1 - D(x)] \right) . \tag{1}$$

We discuss practical amortized methods that avoid the need to train $n$ discriminators in Section 6, but to keep the derivation in this section simple, we consider independent discriminators for now.

## 4.2 Exemplar Models as Implicit Density Estimation

To show how the exemplar model can be used for implicit density estimation, we begin by considering an infinitely powerful, optimal discriminator, for which we can make an explicit connection between the discriminator and the underlying data distribution $P_{\mathcal{X}}(x)$:

**Proposition 1.** *(Optimal Discriminator) For a discrete distribution $P_{\mathcal{X}}(x)$, the optimal discriminator $D_{x^*}$ for exemplar $x^*$ satisfies*

$$D_{x^*}(x) = \frac{\delta_{x^*}(x)}{\delta_{x^*}(x) + P_{\mathcal{X}}(x)} \qquad and \qquad D_{x^*}(x^*) = \frac{1}{1 + P_{\mathcal{X}}(x^*)}.$$

*Proof.* The proof is obtained by taking the derivative of the loss in Eq. (1) with respect to $D(x)$, setting it to zero, and solving for $D(x)$. $\square$

It follows that, if the discriminator is optimal, we can recover the probability of a data point $P_{\mathcal{X}}(x^*)$ by evaluating the discriminator at its own exemplar $x^*$, according to

$$P_{\mathcal{X}}(x^*) = \frac{1 - D_{x^*}(x^*)}{D_{x^*}(x^*)}. \tag{2}$$

For continuous domains, $\delta_{x^*}(x^*) \to \infty$, so $D(x) \to 1$. This means we are unable to recover $P_{\mathcal{X}}(x)$ via Eq. (2). However, we can smooth the delta by adding noise $\epsilon \sim q(\epsilon)$ to the exemplar $x^*$ during training, which allows us to recover exact density estimates by solving for $P_{\mathcal{X}}(x)$. For example, if we let $q = \mathcal{N}(0, \sigma^2 I)$, then the optimal discriminator evaluated at $x^*$ satisfies $D_{x^*}(x^*) = \left[1/\sqrt{2\pi\sigma^2}^d\right] / \left[1/\sqrt{2\pi\sigma^2}^d + P_{\mathcal{X}}(x)\right]$. Even if we do not know the noise variance, we have

$$P_{\mathcal{X}}(x^*) \propto \frac{1 - D_{x^*}(x^*)}{D_{x^*}(x^*)}. \tag{3}$$

This proportionality holds for any noise $q$ as long as $(\delta_{x^*} * q)(x^*)$ (where $*$ denotes convolution) is the same for every $x^*$. The reward bonus we describe in Section 5 is invariant to the normalization factor, so proportional estimates are sufficient.

In practice, we can get density estimates that are better suited for exploration by introducing smoothing, which involves adding noise to the background distribution $P_{\mathcal{X}}$, to produce the estimator

$$D_{x^*}(x) = \frac{(\delta_{x^*} * q)(x)}{(\delta_{x^*} * q)(x) + (P_{\mathcal{X}} * q)(x)}.$$

We then recover our density estimate as $(P_{\mathcal{X}} * q)(x^*)$. In the case when $P_{\mathcal{X}}$ is a collection of delta functions around data points, this is equivalent to kernel density estimation using the noise distribution as a kernel. With Gaussian noise $q = \mathcal{N}(0, \sigma^2 I)$, this is equivalent to using an RBF kernel.

## 4.3 Latent Space Smoothing with Noisy Discriminators

In the previous section, we discussed how adding noise can provide for smoothed density estimates, which is especially important in complex or continuous spaces, where all states might be distinguishable with a powerful enough discriminator. Unfortunately, for high-dimensional states, such as images, adding noise directly to the state often does not produce meaningful new states, since the distribution of states lies on a thin manifold, and any added noise will lift the noisy state off of this manifold. In this section, we discuss how we can learn a smoothing distribution by injecting the noise into a learned latent space, rather than adding it to the original states.

Formally, we introduce a latent variable $z$. We wish to train an encoder distribution $q(z|x)$, and a latent space classifier $p(y|z) = D(z)^y(1 - D(z))^{1-y}$, where $y = 1$ when $x = x^*$ and $y = 0$ when $x \neq x^*$. We additionally regularize the noise distribution against a prior distribution $p(z)$, which in our case is a unit Gaussian. Letting $\widetilde{p}(x) = \frac{1}{2}\delta_{x^*}(x) + \frac{1}{2}p_{\mathcal{X}}(x)$ denote the balanced training distribution from before, we can learn the latent space by maximizing the objective

$$\max_{p_{y|z}, q_{z|x}} E_{\widetilde{p}}[E_{q_{z|x}}[\log p(y|z)] - D_{KL}(q(z|x)||p(z))]. \tag{4}$$

Intuitively, this objective optimizes the noise distribution so as to maximize classification accuracy while transmitting as little information through the latent space as possible. This causes $z$ to only capture the factors of variation in $x$ that are most informative for distinguish points from the exemplar, resulting in noise that stays on the state manifold. For example, in the Atari domain, latent space noise might correspond to smoothing over the location of the player and moving objects on the screen, in contrast to performing pixel-wise Gaussian smoothing.

Letting $q(z|y = 1) = \int_x \delta_{x^*}(x)q(z|x)dx$ and $q(z|y = 0) = \int_x p_{\mathcal{X}}(x)q(z|x)dx$ denote the marginalized positive and negative densities over the latent space, we can characterize the optimal discriminator and encoder distributions as follows. For any encoder $q(z|x)$, the optimal discriminator $D(z)$ satisfies:

$$p(y = 1|z) = D(z) = \frac{q(z|y = 1)}{q(z|y = 1) + q(z|y = 0)},$$

and for any discriminator $D(z)$, the optimal encoder distribution satisfies:

$$q(z|x) \propto D(z)^{y_{\text{soft}}(x)}(1 - D(z))^{1-y_{\text{soft}}(x)}p(z),$$

where $y_{\text{soft}}(x) = p(y = 1|x) = \frac{\delta_{x^*}(x)}{\delta_{x^*}(x) + p_{\mathcal{X}}(x)}$ is the average label of $x$. These can be obtained by differentiating the objective, and the full derivation is included in Appendix A.1. Intuitively, $q(z|x)$ is equal to the prior $p(z)$ by default, which carries no information about $x$. It then scales up the probability on latent codes $z$ where the discriminator is confident and correct. To recover a density estimate, we estimate $D(x) = E_q[D(z)]$ and apply Eq. (3) to obtain the density.

## 4.4 Smoothing from Suboptimal Discriminators

In our previous derivations, we assume an optimal, infinitely powerful discriminator which can emit a different value $D(x)$ for every input $x$. However, this is typically not possible except for small, countable domains. A secondary but important source of density smoothing occurs when the discriminator has difficulty distinguishing two states $x$ and $x'$. In this case, the discriminator will average over the outputs of the infinitely powerful discriminator. This form of smoothing comes from the inductive bias of the discriminator, which is difficult to quantify. In practice, we typically found this effect to be beneficial for our model rather than harmful. An example of such smoothed density estimates is shown in Figure 2. Due to this effect, adding noise is not strictly necessary to benefit from smoothing, though it provides for significantly better control over the degree of smoothing.

## 5 EX$^2$: Exploration with Exemplar Models

We can now describe our exploration algorithm based on implicit density models. Pseudocode for a batch policy search variant using the single exemplar model is shown in Algorithm 1. Online variants for other RL algorithms, such as Q-learning, are also possible. In order to apply the ideas from count-based exploration described in Section 3, we must approximate the state visitation counts $N(s) = nP(s)$, where $P(s)$ is the distribution over states visited during training. Note that we can easily use state-action counts $N(s, a)$, but we omit the action for simplicity of notation. To generate approximate samples from $P(s)$, we use a replay buffer $B$, which is a first-in first-out (FIFO) queue that holds previously visited states. Our exemplars are the states we wish to score, which are the states in the current batch of trajectories. In an online algorithm, we would instead train a discriminator after receiving every new observation one at a time, and compute the bonus in the same manner.

Given the output from discriminators trained to optimize Eq (1), we augment the reward with a function of the "novelty" of the state (where $\beta$ is a hyperparameter that can be tuned to the magnitude of the task reward): $R'(s, a) = R(s, a) + \beta f(D_s(s))$.

**Algorithm 1** EX$^2$ for batch policy optimization

---

1: Initialize replay buffer $B$
2: **for** iteration $i$ in $\{1, \ldots, N\}$ **do**
3:   Sample trajectories $\{\tau_j\}$ from policy $\pi_i$
4:   **for** state $s$ in $\{\tau\}$ **do**
5:     Sample a batch of negatives $\{s'_k\}$ from $B$.
6:     Train discriminator $D_s$ to minimize Eq. (1) with positive $s$, and negatives $\{s'_k\}$.
7:     Compute reward $R'(s,a) = R(s,a) + \beta f(D_s(s))$
8:   **end for**
9:   Improve $\pi_i$ with respect to $R'(s,a)$ using any policy optimization method.
10:   $B \leftarrow B \cup \{\tau_i\}$
11: **end for**

---

In our experiments, we use the heuristic bonus $-\log p(s)$, due to the fact that normalization constants become absorbed by baselines used in typical RL algorithms. For discrete domains, we can also use a count-based $1/\sqrt{N(s)}$ (Tang et al., 2017), where $N(s) = nP(s)$, and $n$ being the size of the replay buffer $B$. A summary of EX$^2$ for a generic batch reinforcement learner is shown in Algorithm 1.

## 6   Model Architecture

To process complex observations such as images, we implement our exemplar model using neural networks, with convolutional models used for image-based domains. To reduce the computational cost of training such large per-exemplar classifiers, we explore two methods for amortizing the computation across multiple exemplars.

### 6.1   Amortized Multi-Exemplar Model

Instead of training a separate classifier for each exemplar, we can instead train a single model that is conditioned on the exemplar $x^*$. When using the latent space formulation, we condition the latent space discriminator $p(y|z)$ on an encoded version of $x^*$ given by $q(z^*|x^*)$, resulting in a classifier for the form $p(y|z, z^*) = D(z, z^*)^y(1 - D(z, z^*))^{1-y}$. The advantage of this amortized model is that it does not require us to train new discriminators from scratch at each iteration, and provides some degree of generalization for density estimation at new states. A diagram of this architecture is shown in Figure 1. The amortized architecture has the appearance of a comparison operator: it is trained to output 0 when $x^* \neq x$, and the optimal discriminator values covered in Section 4 when $x^* = x$, subject to the smoothing imposed by the latent space noise.

### 6.2   K-Exemplar Model

As long as the distribution of positive examples is known, we can recover density estimates via Eq. (3). Thus, we can also consider a batch of exemplars $x_1, ..., x_K$, and sample from this batch uniformly during training. We refer to this model as the "K-Exemplar" model, which allows us to interpolate smoothly between a more powerful model with one discriminator per state ($K = 1$) with a weaker model that uses a single discriminator for all states ($K = \#$ states). A more detailed discussion of this method is included in Appendix A.2. In our experiments, we batch adjacent states in a trajectory into the same discriminator which corresponds to a form of temporal regularization that assumes that adjacent states in time are similar. We also share the majority of layers between discriminators in the neural networks similar to (Osband et al., 2016), and only allow the final linear layer to vary amongst discriminators, which forces the shared layers to learn a joint feature representation, similarly to the amortized model. An example architecture is shown in Figure 1.

### 6.3   Relationship to Generative Adverserial Networks (GANs)

Our exploration algorithm has an interesting interpretation related to GANs (Goodfellow et al., 2014). The policy can be viewed as the generator of a GAN, and the exemplar model serves as the discriminator, which is trying to classify states from the current batch of trajectories against previous

a) Amortized Architecture                    b) K-Exemplar Architecture

Figure 1: A diagram of our a) amortized model architecture and b) the K-exemplar model architecture. Noise is injected after the encoder module (a) or after the shared layers (b). Although possible, we do not tie the encoders of (a) in our experiments.

states. Using the K-exemplar version of our algorithm, we can train a single discriminator for all states in the current batch (rather than one for each state), which mirrors the GAN setup.

In GANs, the generator plays an adversarial game with the discriminator by attempting to produce indistinguishable samples in order to fool the discriminator. However, in our algorithm, the generator is rewarded for helping the discriminator rather than fooling it, so our algorithm plays a cooperative game instead of an adversarial one. Instead, they are competing with the progression of time: as a novel state becomes visited frequently, the replay buffer will become saturated with that state and it will lose its novelty. This property is desirable in that it forces the policy to continually seek new states from which to receive exploration bonuses.

# 7  Experimental Evaluation

The goal of our experimental evaluation is to compare the $EX^2$ method to both a naïve exploration strategy and to recently proposed exploration schemes for deep reinforcement learning based on explicit density modeling. We present results on both low-dimensional benchmark tasks used in prior work, and on more complex vision-based tasks, where prior density-based exploration bonus methods are difficult to apply. We use TRPO (Schulman et al., 2015) for policy optimization, because it operates on both continuous and discrete action spaces, and due to its relative robustness to hyper-parameter choices (Duan et al., 2016). Our code and additional supplementary material including videos will be available at `https://sites.google.com/view/ex2exploration`.

**Experimental Tasks**   Our experiments include three low-dimensional tasks intended to assess whether $EX^2$ can successfully perform implicit density estimation and computer exploration bonuses, and four high-dimensional image-based tasks of varying difficulty intended to evaluate whether implicit density estimation provides improvement in domains where generative modeling is difficult. The first low-dimensional task is a continuous 2D maze with a sparse reward function that only provides a reward when the agent is within a small radius of the goal. Because this task is 2D, we can use it to directly visualize the state visitation densities and compare to an upper bound histogram method for density estimation. The other two low-dimensional tasks are benchmark tasks from the OpenAI gym benchmark suite, SparseHalfCheetah and SwimmerGather, which provide for a comparison against prior work on generative exploration bonuses in the presence of sparse rewards.

For the vision-based tasks, we include three Atari games, as well as a much more difficult ego-centric navigation task based on vizDoom (DoomMyWayHome+). The Atari games are included for easy comparison with prior methods based on generative models, but do not provide especially challenging visual observations, since the clean 2D visuals and relatively low visual diversity of these tasks makes generative modeling easy. In fact, prior work on video prediction for Atari games easily achieves accurate predictions hundreds of frames into the future (Oh et al., 2015), while video prediction on natural images is challenging even a couple of frames into the future (Mathieu et al., 2015). The vizDoom maze navigation task is intended to provide a comparison against prior methods with substantially more challenging observations: the game features a first-person viewpoint, 3D visuals, and partial observability, as well as the usual challenges associated with sparse rewards. We make the task particularly difficult by initializing the agent in the furthest room from the goal location,

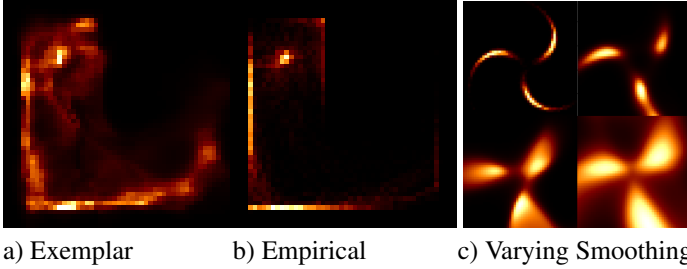

a) Exemplar          b) Empirical          c) Varying Smoothing

Figure 2: **a, b**) Illustration of estimated densities on the 2D maze task produced by our model (a), compared to the empirical discretized distribution (b). Our method provides reasonable, somewhat smoothed density estimates. **c**) Density estimates produced with our implicit density estimator on a toy dataset (top left), with increasing amounts of noise regularization.

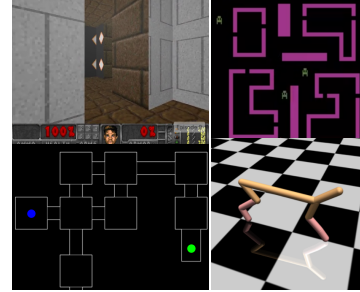

Figure 3: Example task images. From top to bottom, left to right: Doom, map of the MyWayHome task (goal is green, start is blue), Venture, HalfCheetah.

requiring it to navigate through 8 rooms before reaching the goal. Sample images taken from several of these tasks are shown in Figure 3 and detailed task descriptions are given in Appendix A.3.

We compare the two variants of our method (K-exemplar and amortized) to standard random exploration, kernel density estimation (KDE) with RBF kernels, a method based on Bayesian neural network generative models called VIME (Houthooft et al., 2016), and exploration bonuses based on hashing of latent spaces learned via an autoencoder (Tang et al., 2017).

**2D Maze**   On the 2D maze task, we can visually compare the estimated state density from our exemplar model and the empirical state-visitation distribution sampled from the replay buffer, as shown in Figure 2. Our model generates sensible density estimates that smooth out the true empirical distribution. For exploration performance, shown in Table 1,TRPO with Gaussian exploration cannot find the sparse reward goal, while both variants of our method perform similarly to VIME and KDE. Since the dimensionality of the task is low, we also use a histogram-based method to estimate the density, which provides an upper bound on the performance of count-based exploration on this task.

**Continuous Control: SwimmerGather and SparseHalfCheetah**   SwimmerGather and Sparse-HalfCheetah are two challenging continuous control tasks proposed by Houthooft et al. (2016). Both environments feature sparse reward and medium-dimensional observations (33 and 20 dimensions respectively). SwimmerGather is a hierarchical task in which no previous algorithms using naïve exploration have made any progress. Our results demonstrate that, even on medium-dimensional tasks where explicit generative models should perform well, our implicit density estimation approach achieves competitive results. $EX^2$, VIME, and Hashing significantly outperform the naïve TRPO algorithm and KDE on SwimmerGather, and amortized $EX^2$outperforms all other methods on Sparse-HalfCheetah by a significant margin. This indicates that the implicit density estimates obtained by our method provide for exploration bonuses that are competitive with a variety of explicit density estimation techniques.

**Image-Based Control: Atari and Doom**   In our final set of experiments, we test the ability of our algorithm to scale to rich sensory inputs and high dimensional image-based state spaces. We chose several Atari games that have sparse rewards and present an exploration challenge, as well as a maze navigation benchmark based on vizDoom. Each domain presents a unique set of challenges. The vizDoom domain contains the most realistic images, and the environment is viewed from an egocentric perspective which makes building dynamics models difficult and increases the importance of intelligent smoothing and generalization. The Atari games (Freeway, Frostbite, Venture) contain simpler images from a third-person viewpoint, but often contain many moving, distractor objects that a density model must generalize to. Freeway and Venture contain sparse reward, and Frostbite contains a small amount of dense reward but attaining higher scores typically requires exploration.

Our results demonstrate that $EX^2$ is able to generate coherent exploration behavior even high-dimensional visual environments, matching the best-performing prior methods on the Atari games. On the most challenging task, DoomMyWayHome+, our method greatly exceeds all of the prior

| Task | K-Ex.(ours) | Amor.(ours) | VIME[1] | TRPO[2] | Hashing[3] | KDE | Histogram |
|------|------------|-------------|---------|---------|-----------|-----|-----------|
| 2D Maze | -104.2 | -132.2 | -135.5 | -175.6 | - | -117.5 | -69.6 |
| SparseHalfCheetah | 3.56 | 173.2 | 98.0 | 0 | 0.5 | 0 | - |
| SwimmerGather | 0.228 | 0.240 | 0.196 | 0 | 0.258 | 0.098 | - |
| Freeway (Atari) | - | 33.3 | - | 16.5 | 33.5 | - | - |
| Frostbite (Atari) | - | 4901 | - | 2869 | 5214 | - | - |
| Venture (Atari) | - | 900 | - | 121 | 445 | - | - |
| DoomMyWayHome | 0.740 | 0.788 | 0.443 | 0.250 | 0.331 | 0.195 | - |

[1] Houthooft et al. (2016)  [2] Schulman et al. (2015)  [3] Tang et al. (2017)

Table 1: Mean scores (higher is better) of our algorithm (both K-exemplar and amortized) versus VIME (Houthooft et al., 2016), baseline TRPO, Hashing, and kernel density estimation (KDE). Our approach generally matches the performance of previous explicit density estimation methods, and greatly exceeds their performance on the challenging DoomMyWayHome+ task, which features camera motion, partial observability, and extremely sparse rewards. We did not run VIME or K-Exemplar on Atari games due to computational cost. Atari games are trained for 50 M time steps. Learning curves are included in Appendix A.5

exploration techniques, and is able to guide the agent through multiple rooms to the goal. This result indicates the benefit of implicit density estimation: while explicit density estimators can achieve good results on simple, clean images in the Atari games, they begin to struggle with the more complex egocentric observations in vizDoom, while our $EX^2$ is able to provide reasonable density estimates and achieves good results.

# 8 Conclusion and Future Work

We presented $EX^2$, a scalable exploration strategy based on training discriminative exemplar models to assign novelty bonuses. We also demonstrate a novel connection between exemplar models and density estimation, which motivates our algorithm as approximating pseudo-count exploration. This density estimation technique also does not require reconstructing samples to train, unlike most methods for training generative or energy-based models. Our empirical results show that $EX^2$ tends to achieve comparable results to the previous state-of-the-art for continuous control tasks on low-dimensional environments, and can scale gracefully to handle rich sensory inputs such as images. Since our method avoids the need for generative modeling of complex image-based observations, it exceeds the performance of prior generative methods on domains with more complex observation functions, such as the egocentric Doom navigation task.

To understand the tradeoffs between discriminatively trained exemplar models and generative modeling, it helps to consider the behavior of the two methods when overfitting or underfitting. Both methods will assign flat bonuses when underfitting and high bonuses to all new states when overfitting. However, in the case of exemplar models, overfitting is easy with high dimensional observations, especially in the amortized model where the network simply acts as a comparator. Underfitting is also easy to achieve, simply by increasing the magnitude of the noise injected into the latent space. Therefore, although both approach can suffer from overfitting and underfitting, the exemplar method provides a single hyperparameter that interpolates between these extremes without changing the model. An exciting avenue for future work would be to adjust this smoothing factor automatically, based on the amount of available data. More generally, implicit density estimation with exemplar models is likely to be of use in other density estimation applications, and exploring such applications would another exciting direction for future work.

**Acknowledgement**   We would like to thank Adam Stooke, Sandy Huang, and Haoran Tang for providing efficient and parallelizable policy search code. We thank Joshua Achiam for help with setting up benchmark tasks. This research was supported by NSF IIS-1614653, NSF IIS-1700696, an ONR Young Investigator Program award, and Berkeley DeepDrive.

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
