[Supplementary Material]

# A  Appendix

## A.1  Noisy Discriminators

Our noisy latent space discriminator of Section 4.3 optimizes the objective:

$$\max_{p_{y|z}, q_{z|x}} E_{\widetilde{p}}[E_{q_{z|x}}[\log p(y|z)] - D_{KL}(q(z|x)||p(z))] \tag{5}$$

Where $\widetilde{p}(x)$ is a balanced dataset of positives $x \sim \delta_{x^*}(x)$ with label $y = 1$, and negatives $x \sim p_{\mathcal{X}}(x)$ with label $y = 0$.

**Proposition 2.** *(Noisy Optimal Discriminator) For any encoder distribution $q(z|x)$, the optimal noisy discriminator of Section 4.3 satisfies*

$$D(z) = \frac{q(z|y=1)}{q(z|y=1) + q(z|y=0)}.$$

*Proof.* This is readily obtained by differentiating the objective with respect to $D(z)$. First we rewrite Eq. (5) in terms of $D(z)$:

$$\mathcal{L} = E_{x,y \sim \widetilde{p}}[\int_z q(z|x)(y \log D(z) + (1-y)\log(1-D(z))] - D_{KL}(q(z|x)||p(z))$$

Differentiating and setting to 0, we obtain:

$$\frac{\partial \mathcal{L}}{\partial D(z)} = \int_{x,y} \widetilde{p}(x,y)q(z|x)(y\frac{1}{D(z)} - (1-y)\frac{1}{1-D(z)})d\{x,y\} = 0$$

Splitting up the positive $\widetilde{p}(x|y=1)$ and negative $\widetilde{p}(x|y=0)$ distributions, we have:

$$\frac{1}{2}\frac{1}{D(z)}\underbrace{\int_x \delta_{x^*}(x)q(z|x)dx}_{q(z|y=1)} - \frac{1}{2}\frac{1}{1-D(z)}\underbrace{\int_x p_{\mathcal{X}}(x)q(z|x)dx}_{q(z|y=0)} = 0$$

Solving for $D(z)$ yields the desired result. □

We can also write down the form of the optimal encoder to understand how the objective shapes the encoding distribution:

**Proposition 3.** *(Noisy Optimal Encoder) For any discriminator $D(z)$, the optimal encoder of Section 4.3 satisfies*

$$q(z|x) \propto D(z)^{y_{soft}(x)}(1 - D(z))^{1-y_{soft}(x)}p(z).$$

*Where $y_{soft}(x) = p(y=1|x) = \frac{\delta_{x^*}(x)}{\delta_{x^*}(x) + p_{\mathcal{X}}(x)}$ is the average label of $x$.*

*Proof.* This is readily obtained by differentiating the objective with respect to $q(z|x)$. Letting $\mathcal{L}$ denote the objective of Eq. (5):

$$0 = \frac{\partial \mathcal{L}}{\partial q(z|x)} = \frac{\partial}{\partial q(z|x)}\int_{y,x} p(y|x)\widetilde{p}(x)[\int_z q(z|x)\log p(y|z)dz - \int_z q(z|x)\log\frac{q(z|x)}{p(z)}dz]d\{x,y\}$$

$$0 = \int_y p(y|x)[\log p(y|z) - 1 - \log q(z|x) + \log p(z)]dy$$

Rearranging,

$$\log q(z|x) = 1 + \log p(z) + \int_y p(y|x)\log p(y|z)dy$$

$$q(z|x) \propto p(z)e^{\int_y p(y|x)\log p(y|z)dy} = p(z)[D(z)^{p(y=1|x)}(1 - D(z))^{p(y=0|x)}]$$

□

| a) 2D Maze | b) SparseHalfCheetah | c) DoomMyWayHome+ |

| d) SwimmerGather | e) Freeway | f) Frostbite | g) Venture |

Figure 4: Illustrations of several tasks used in our experiments.

## A.2 K-Exemplar Model

In the *K-exemplar model*, each discriminator is associated with a batch of K positive exemplars $B = \{x_1, \ldots x_K\}$. In this case, we sample positives from the batch $B$ uniformly at random rather than always using a single exemplar. Letting $P_B(x)$ denote a uniform distribution over $B$, we optimize

$$D_B = \arg\max_{D \in \mathcal{D}} \left( E_{x \sim P_B}[\log D(x)] + E_{x' \sim P_{\mathcal{X}}}[\log 1 - D(x')] \right). \tag{6}$$

Using the same argument as the single exemplar model, we can characterize the optimal discriminator for the noiseless $K$-exemplar model:

**Proposition 4.** *(K-Exemplar Optimal Discriminator) For a discriminator trained with K positives $\{x_1, \ldots x_K\}$ sampled uniformly, the optimal discriminator $D_B^*$ evaluated at any one of the positives $x$ satisfies*

$$D_B^*(x) = \frac{1}{1 + K P_{\mathcal{X}}(x)}.$$

*Proof.* Taking the derivative of Equation (6) with respect to $D_B^*(x)$, we obtain

$$\frac{1}{K D_B^*(x)} - \frac{P(x)}{1 - D_B^*(x)} = 0.$$

Solving for $D_B^*(x)$ yields the desired result. $\qquad\square$

Extensions to noisy versions of the K-exemplar model follow in exactly the same way as the single exemplar model, only changing the positive distribution from $\delta_{x^*}(x)$ to $P_B(x)$.

## A.3 Task Descriptions

In this section we describe the tasks used in our experiments. Sample images from these tasks are included in Figure 4.

**2D Maze.** This task involves navigating through a 2D maze, using the (x,y) coordinate of the agent as the observation. The challenge stems from the sparse reward, which is only obtained in a small box around the goal. The agent therefore has to figure out how to reach novel parts of the maze in order to eventually find the reward region.

**SparseHalfCheetah.** This task involves making a 6-DoF robot run forward as fast as possible. However, this task has been modified to have a sparse reward as done by Houthooft et al. (2016), so

Figure 5: Bonuses assigned by exemplar model during the middle of training, plotted by XY position in the maze (left) shown with the maze (right) for reference. Yellow/white denotes high bonus, red denotes low bonus. Note that the starting room (with the blue dot) is assigned low bonus, and the outer rooms are assigned higher bonuses.

that the agent only receives reward upon reaching a certain position threshold, and receives a constant reward afterwards.

**SwimmerGather.** This locomotion task, initially proposed as a hierarchical task by Duan et al. (2016), involves navigating a 3-link snake-like robot to collect green or red pellets. The agent is rewarded for collecting green pellets and penalized for red ones.

**Doom (MyWayHome+).** This task involves navigating an agent through a maze to find a vest that is located in one of the rooms. The observations consist only of visual feedback, and the reward is sparse and only given when the vest is obtained. This is a slightly modified version of the OpenAI Gym task where we initialize the agent in the furthest room from the vest to create a sparse reward task. In Figure 4, the map of the environment is shown, with the agent starting at the blue dot and the goal at the green dot. The input is resized to an RGB 32 x 32 image. A sample map of the bonuses along trajectories during the middle of training is shown in Figure 5.

**Freeway.** This game involves navigating an agent across a highway with moving cars, which push the agent back when touched. The reward is sparse and the agent scores a 1 when it makes it across the highway.

**Frostbite.** This game involves an agent jumping across ice platforms floating across a river. The reward is dense in that the agent receives reward when it jumps on a platform, but higher scores requires the agent to navigate to other stages which generally requires exploration.

**Venture.** This game involves an agent navigating an agent into multiple rooms, where reward is received upon picking up certain objects. The agent must avoid death from touching wandering enemies. We show example images with low and high bonuses given by our algorithm on this task in Figure 6.

## A.4 Experiment Hyperparameters

### A.4.1 Policy Model Parameters

We used an identical fully connected policy architecture across all non-image tasks, and a convolutional architecture for the image task.

For non-image tasks, we used a 2-layer neural network with 32 hidden units per layer, and relu nonlinearities.

Figure 6: Top: 3 of the lowest scoring images on Venture early during training. These are typically pictures of the agent in the "overworld" where it spends most of its time. Bottom: 3 of the highest scoring images, which are typically when the agent enters one of the many rooms with reward. Images are grayscale due to preprocessing of the image.

For Doom, we used 2 convolutional layers (16 4x4 filters, stride 2) followed by 2 fully connected layers with 32 units each. All nonlinearities were relus. We resize the input screen to a RGB 32 x 32 image. For Atari, we used 2 convolutional layers (32 8x8 filters, stride 4, 16 4x4 filter stride 2) followed by 2 fully connected layers with 256 units each. All nonlinearities were relus. For Atari we use the last 4 frames each resized to a grayscale 42 x 42 image.

### A.4.2 Exemplar Model Parameters

We used an identical fully connected exemplar architecture across all non-image tasks, and a convolutional architecture for the image task.

For non-image tasks, we used a 2-layer shared neural network with $\tanh$ nonlinearities and 16 units per layer. The final unshared layer was a linear layer.

For image-based tasks, we used a shared network consisting of 2 convolutional layers (16 4x4 filters, stride 2) followed by 2 fully connected layers with 16 units each. The convolutional layers used relu nonlinearities, and the fully connected used $\tanh$. The shared network architecture is identical to the policy architecture. The final unshared layer was a linear layer.

We also found it useful to lower the learning rate for the shared network as it has many more gradients backpropogating through it than the unshared layer. Thus, we optimized our model using ADAM with a learning rate of $5 * 10^{-4}$ for the shared layers and $1 * 10^{-3}$ for the unshared layers.

### A.4.3 Amortized Model Parameters

For each encoder we use a 2-layer neural network with 32 hidden units per layer and tanh nonlinearities which outputs the mean and log variance of the latent representation of size 16. The latent codes of the encoder are concatenated and fed into the discriminator which is another 2-layer neural network with 32 hidden units per layer and tanh nonlinearities.

For image-based tasks, we preprocess the input with 2 convolutional layers (16 4x4 filters, stride 2) before feeding the input into the encoders. For the encoders and discriminator we use the same architecture as stated above except we use 64 hidden units and a latent size of 32.

We use a learning rate of $1 * 10^{-4}$ and optimize the model with ADAM. We found it important to tune the weight on the KL divergence loss which affects how well the discriminator can over or under fit.

### A.4.4 Task Specific EX$^2$ Parameters

We found it best to tune the exploration bonus weight $\beta$ to match the magnitude of the reward of the task. We used the following EX$^2$ hyperparameters for each task, which were obtained via a rough grid search over possible values:

**2D Maze.** We use K-Exemplar (K=5) and an exploration bonus weight of 1.0. For the amortized model we use an exploration bonus weight of 0.01 and KL divergence weight of 0.01.

**HalfCheetah.** We use K-Exemplar (K=5) and an exploration bonus weight of 0.001. For the amortized model we use an exploration bonus weight of 0.001 and KL divergence weight of 0.1.

**SwimmerGather.** We use single exemplars with an exploration bonus weight of 1.0. For the amortized model we use an exploration bonus weight of $1 * 10^{-4}$ and KL divergence weight of 10.

**Doom (MyWayHome).** We use K-Exemplar (K=5), an exploration bonus weight of $1 * 10^{-4}$, and entropy bonus of $1 * 10^{-5}$. For the amortized model we use an exploration bonus weight of $1 * 10^{-4}$ and KL divergence weight of 0.01.

**Freeway** For the amortized model we use an exploration bonus weight of $1 * 10^{-5}$ and KL divergence weight of 0.1.

**Frostbite** For the amortized model we use an exploration bonus weight of 0.001 and KL divergence weight of 0.1.

**Venture** For the amortized model we use an exploration bonus weight of $1 * 10^{-4}$ and KL divergence weight of 0.001.

## A.5 Learning Curves

Figure 7: 2D Maze

Figure 8: Swimmer Gather

Figure 9: SparseHalfCheetah

Figure 10: DoomMyWayHome+

Figure 11: Freeway

Figure 12: Frostbite

Figure 13: Venture