[Reviews · NeurIPS 2017]

Reviewer 1



I really enjoyed the paper. It tackles the main limitation of the count-based exploration with a novel idea, i.e., by making it possible to get the density without generative modeling. This is done by implicitly modeling the observation density from a discriminator. It is based on the intuition that if the current state has not been visited in the past, it should be easy for a discriminator to distinguish it from other states visited in the past (and thus getting high reward). To make this idea based on the exemplar model practical, the authors propose to learn the noise distribution in latent space and discriminator sharing. Regarding the possibility of the powerful discriminator memorizing perfectly (and thus classifying perfectly in continuous domain), it would be interesting to discuss in relation to the "rethinking generalization" paper (ICLR17). Discussion and experiment on the storage complexity would be helpful, e.g., in the case where the replay buffer size is limited. A related work by D. Pathak et. al. (ICML 17) which takes the prediction error in the latent space as the exploration reward seems to suffer less from the storage complexity. It would also be interesting to see if the authors provide some coverage heatmap for VizDoom experiment.

Reviewer 2



This paper presents EX2, a method for novelty-based exploration. Unlike previous methods, EX2 does not require prediction of future observations, and instead relies on a discriminative model to classify which states are novel (exemplars) and then provide reward bonuses to the visitation of novel states. Results show that EX2 outperforms competing novelty-based exploration methods on a VizDoom navigation task and performs as well as others on a variety of simpler, less graphically intense tasks. I believe this paper makes a novel algorithmic contribution to the body of intelligent exploration literature and has solid experimental results to back up the claims. The paper is well written, experimentally thorough, and a pleasure to read. After reading the rebuttal and other reviews, I still think the paper should be accepted.

Reviewer 3



Review of submission 1489: EX2: Exploration with Exemplar Models for Deep Reinforcement Learning Summary: A discriminative novelty detection algorithm is proposed to improve exploration for policy gradient based reinforcement learning algorithms. The implicitly-estimated density by the discriminative novelty detection of a state is then used to produce a reward bonus added to the original reward for down-stream policy optimization algorithms (TRPO). Two techniques are discussed to improve the computation efficiency. Comments - One motivation of the paper is to utilize implicit density estimation to approximate classic count based exploration. The discriminative novelty detection only maintains a density estimation over the states, but not state-action pairs. To the best of my knowledge, most theoretical results on explorations are built on the state-action visit count, not state visit count. Generally, exploration strategies for RL algorithms investigate reducing the RL agent’s uncertainty over the MDP’s state transition and reward functions. The agent’s uncertainty (in terms of confidence intervals or posterior distributions over environment parameters) decreases as the inverse square root of the state-action visit count. - It is not clear from the texts how the added reward bonus is sufficient to guarantee the novel state action pairs to be visited for policy gradient based RL methods. Similar reward bonuses have been shown to improve exploration for value-based reinforcement learning methods, such as MBIE-EB (Strehl & Littman, 2009) and BEB (Kolter & Ng, 2009), but not policy gradient based RL methods. Value-based and policy gradient based RL methods differ a lot on the technical details. - Even we assume the added reward bonuses could result in novel state action pairs to be visited as desired, it is not clear why policy gradient methods could benefit from such exploration strategy. The paper, State-Dependent Exploration for Policy Gradient Methods (Thomas Ruckstieß, Martin Felder, and Jurgen Schmidhuber, ECML 2008), showed that limiting the exploration within an episode (i.e., returning the same action for any given state within an episode) could be helpful for policy gradient RL methods by reducing the variance in the gradients and improving the credit assignment. The soundness of the proposed techniques would be improved if necessary explanations were provided. - There is one closely related recent paper on improving exploration for policy gradient methods, IMPROVING POLICY GRADIENT BY EXPLORING UNDER-APPRECIATED REWARDS (Ofir Nachum et al. ICLR 2017). It would be good for the paper to discuss the relationship and even compare against it empirically.